# Structural and Vibrational Investigations of Mixtures of Cocoa Butter (CB), Cocoa Butter Equivalent (CBE) and Anhydrous Milk Fat (AMF) to Understand Fat Bloom Process

Mustapha El Hadri [1], Serge Bresson [2,*], Albane Lecuelle [2], Fatiha Bougrioua [3], Vincent Baeten [4], Van Hung Nguyen [5], Vincent Faivre [5] and Matthieu Courty [6]

1 Condensed Matter Physics Team, Abdelmalek Essaâdi University, Tetouan 93000, Morocco; elhadri.mustafa@gmail.com
2 Institut Polytechnique UniLaSalle, Université d'Artois, ULR 7519, 19 Rue Pierre Waguet, BP 30313, 60026 Beauvais, France; albane.lecuelle@gmail.com
3 Departement de Physique, Faculté des Sciences, Université de Picardie Jules Verne, 33 Rue St Leu, 80039 Amiens, France; fatiha.bougrioua@u-picardie.fr
4 Quality and Authentication of Products Unit (U15), Valorisation of Agricultural Products Department, Walloon Agricultural Research Centre (CRA-W), 'Henseval Building', Chaussée de Namur 24, 5030 Gembloux, Belgium; v.baeten@cra.wallonie.be
5 Institut Galien Paris-Saclay, Université Paris-Saclay, 5 Rue JB Clément, 92296 Châtenay-Malabry, France; van-hung.nguyen@universite-paris-saclay.fr (V.H.N.); vincent.faivre@universite-paris-saclay.fr (V.F.)
6 Laboratoire de Réactivité et Chimie des Solides, Université de Picardie Jules Verne, Hub de l'Energie, 15 Rue Baudelocque, 80039 Amiens, France; matthieu.courty@u-picardie.fr
* Correspondence: sergebresson@yahoo.fr

**Abstract:** Some studies found that the proportions of cocoa butter (CB), cocoa butter equivalent (CBE) and milk fatty acid (AMF) tend to influence the blooming delay when mixing them. The goal of our research is to determine the effects of the proportion of CB, CBE and AMF on the structural organization of the final mixtures. X-ray, DSC, MIR and Raman spectroscopy were used to analyze the structural features and the vibrational modes of four mixtures: CB + 0.5AMF, CB + AMF, CB + 0.5AMF + CBE and CB + AMF + CBE. At room temperature, the triglycerides are ingredients of CB, and CBE and AMF do not fully exhibit the known crystalline forms V or VI, unlike a recent CB sample. Part of these triglycerides is in the form IV instead. The presence of the latter seems to be a key parameter that favors the deceleration of the transformation to the form VI, which is responsible for the development of fat bloom.

**Keywords:** cocoa butter (CB); cocoa butter equivalent (CBE); milk fatty acid (AMF); MIR and Raman spectroscopy; X-ray diffraction

## 1. Introduction

The fatty blooming of chocolate is characterized by the loss of the initial gloss of its surface, giving it a more or less white appearance. For milk chocolate, this phenomenon is directly linked to the physico-chemical structure of cocoa butter (CB), cocoa butter equivalent (CBE) and white milk fatty acids (anhydrous milk fat: AMF) [1–6].

Sonvaï and Rousseau [1] have shown that the proportion of triglycerides in cocoa butter (CB), cocoa butter equivalent (CBE) and anhydrous milk fats (AMF) of chocolate strongly influences the speed of appearance of blooming. These authors have shown that the time in weeks before the appearance of polymorphic crystals in form VI responsible for the blooming of chocolate depended on the proportion by mass of the product of cocoa butter (CB), cocoa butter equivalent (CBE) and milk fatty acids (AMF): for the mixtures CB + 0.5AMF and CB + AMF, the blooming of chocolate appears after two weeks, the mixture CB + 0.5AMF + CBE after 20 weeks while for the mixture CB + AMF + CBE it appears after 30 weeks.

We notice that products containing cocoa butter equivalent (CBE) take much longer to bloom than products not containing it. "CB + AMF" mixture, not containing CBE, only takes two weeks before blooming, compared to thirty weeks (i.e., fifteen times more) for "CB + AMF + CBE" mixture or to twenty weeks for "CB + 0.5AMF + CBE" mixture. CBE, therefore, appears to delay very significantly the transition of cocoa butter crystals in V-form to VI-form. In addition, it can be noted that a suitable amount of milk fatty acids (AMF) in a product containing CBE further delays the appearance of crystals in form VI. Indeed, "CB + AMF + CBE" mixture containing twice as much milk fatty acids as "CB + 0.5AMF + CBE" mixture takes 10 more weeks to bloom, that is to say, an additional delay in the transition of 50%. The more milk "chocolate" contains milk powder, in the presence of equivalent cocoa butter, the longer it would take to bloom. Adding fatty acid to milk to delay blooming has also been shown for dark chocolate. Indeed, a few years earlier, it had already been demonstrated that the addition of 1 to 2% of milk fatty acid in dark chocolate delayed the blooming of the product [7]. This could be explained by the presence of triglycerides in the fatty acids of milk but not in cocoa butter. Another, more recent, study, conducted by Bisvas et al. [8], also demonstrated that the presence of cocoa butter substitute (from palm oil, noted CBS) in the composition of a dark chocolate delayed the blooming of the product. Indeed, they showed that after two weeks of storage at 29 °C ± 1 °C, a dark chocolate containing no cocoa butter substitute (CBS) bloomed, unlike a dark chocolate containing 20% CBS. However, their experiments also showed that chocolate containing only 5% CBS bloomed just like chocolate not containing CBS. The presence of CBS in dark chocolate therefore makes it possible to delay blooming, provided that its proportion in the product is sufficient. Da Silva et al. [9] also conclude that when the chocolate was subjected to temperature cycling, the resistance of CBS and CBE to the formation of fat bloom became more evident.

ATR-FTIR and Raman spectroscopy are non-destructive vibrational spectroscopy techniques which give sensitive information about molecular structure in solid and liquid TG conformational dependence [6,10–12]. Some bands of Raman spectra are particularly interesting to investigate the polymorphic structure of AMF. The bands in the spectral region 3200–2700 cm$^{-1}$ correspond to the $\nu$(C–H) stretching modes. The $\nu$(C=O) ester carbonyl stretching region appears at 1800–1700 cm$^{-1}$, and the $\nu$(C=C) stretching region (olefinic band) near 1660 cm$^{-1}$.

After separately studying the cocoa butter, the equivalent cocoa butter and the fatty acids of milk [13–15], we present a structural and vibrational study of the four mixtures studied by Sonvaï and Rousseau by X-ray diffraction, DSC, MIR and Raman spectroscopies. The objective is to better understand the phenomenon of fatty blooming of chocolate observed by Sonvaï and Rousseau for the same four mixtures. For this, we looked for markers of differentiation between these samples by structural studies according to the temperature and by vibrational studies.

## 2. Materials and Methods

### 2.1. Mixtures Preparation

Cocoa butter (CB) came from the Ivory Coast. CB, CBE and AMF were both purchased from the industry Cadbury (Canada). From mass spectrometry experiments by two different techniques, ESI-HRMS and MALDI-HRMS [6], we can conclude that our samples of CB and CBE are identical with respect to the types of triglycerides in them, but regarding the three main triglycerides, the components POS, SOS and POP, the proportions are different: in CB, POS dominates in quantity compared to the other two (practically 50% of the total), while for CBE, POP plays this role (practically 46% of the total) with a slight increase of SOS.

Cocoa butter, cocoa butter equivalent and milk fatty acids are solid fats at room temperature. They therefore had to be melted to make the mixtures at T = 60 °C. Then, the samples were stored in a refrigerator at T = 4 °C until used for the experiments at room temperature. Using a weighing scale accurate to one hundredth of a gram, the required mass of each sample was weighed. The samples were then melted using a heating magnetic

stirrer, then mixed using the magnetic stirrer to obtain a homogeneous liquid. The cooling then took place naturally in the open air. In Table 1, the mixtures are presented by mass.

**Table 1.** Composition of the mixtures of cocoa butter (CB), cocoa butter equivalent (CBE) and milk fatty acids (AMF).

| Mixture | Mass of CB | Mass of AMF | Mass of CBE |
|---|---|---|---|
| "CB + 0.5AMF" | 10.00 g | 5.00 g | x |
| "CB + AMF" | 10.00 g | 10.00 g | x |
| "CB + 0.5AMF + CBE" | 10.00 g | 5.00 g | 10.00 g |
| "CB + AMF + CBE" | 10.00 g | 10.00 g | 10.00 g |

### 2.2. SAXS-WAXS Experiments

X-ray diffraction patterns were acquired using a microfocus X-ray tube (IµS, Incoatec, Geesthacht, Germany), selecting the Cu Kα radiation. It was used with an intensity of 1000 µA and a voltage of 50 kV. The incident beam was focused at the detector with multilayer Montel optics and 2D Kratky block collimator. Small-angle (SAXS) and wide-angle (WAXS) X-ray scattering analyses were performed simultaneously using two position-sensitive linear detectors (Vantec-1, Bruker, Billerica, MA, USA) set perpendicular to the incident beam direction, up to 7° (2θ) and at 19° to 28° (2θ) from it, respectively. The direct beam was stopped with a W-filter. The scattered intensity was reported as a function of the scattering vector $q = 4\pi \sin\theta/\lambda$ where $\theta$ is half the scattering angle and $\lambda$ is the wavelength of the radiation. The repeat distances d, characteristic of the structural arrangements, were given by $q$ (Å$^{-1}$) = $2\pi/d$ (Å). Silver behenate and tristearin (β form) were used as standards to calibrate SAXS and WAXS detectors, respectively.

All samples (~10 mg) were introduced into thin-walled glass capillaries (GLAS, Müller, Berlin, Germany) of 1.5 mm external diameter which were then placed in a specially designed temperature-controlled sample holder (Microcalix, Setaram, Lyon, France). For static measurements at 20 °C, the acquisition time was 10 min. For measurements with temperature, samples were heated at 2 °C/min and acquisition time was 1 min, leading to frame recording every 2 °C.

### 2.3. Differential Scanning Calorimetry (DSC)

DSC experiments were carried out on a Netzsch DSC 204 F1 Phoenix® heat flux differential calorimeter at a heating rate of 2 °C/min under a constant argon flow with 200 mL/min. Each sample was heated from room temperature to T = 60 °C. Samples were weighed in aluminum sample pans covered with a pierced lid. An empty aluminum sample pan with a pierced lid was used as a reference. Three temperatures could be measured: $T_{onset}$, $T_{max}$ and $T_{offset}$, which correspond respectively to the beginning, the top and the end of thermal events.

### 2.4. MIR Spectroscopy

The MIR measurements were carried out at the Walloon Agricultural Research Center CRA-W, in Belgium. The apparatus consists of an FT-MIR Vertex 70 spectrometer (Brukeroptics, Ettlingen, Germany) equipped with a Golden Gate ATR (Attenuated Total Reflectance). This ATR consists of a monolithic diamond crystal. The incident beam contains radiations of 4000 to 600 cm$^{-1}$, which correspond to the medium infrared. The incident light penetrates the sample to a depth of 7 µm maximum, after having passed through the diamond. The reflected beam emerges through the diamond before reaching the detector. The spectral resolution was set at 1 cm$^{-1}$ and the number of co-added spectra was set to 128 scans. The measurements were carried out at room temperature. Spectra of the ambient air was used as background. The ATR-FTIR spectra had undergone special processing, in order to be able to compare the intensity ratios between the samples. Before

normalizing the spectra on the peak at 1729 cm$^{-1}$, which corresponded to the most intense peak in the spectral region 2000–50 cm$^{-1}$, we subtracted the baseline for each spectrum.

### 2.5. Raman Spectroscopy

The measurements were carried out at the Walloon Agricultural Research Center (CRA-W), Belgium.

RAMAN spectra were acquired using a SENTERRA II Bruker RAMAN spectrometer. This fully automated instrument combines excellent sensitivity and high resolution of 1.5 cm$^{-1}$. The experiments were carried out with a laser of wavelength $\lambda_0$ = 532 nm, of maximum power, Pmax = 25 mW, an acquisition time of 100 s and an addition of two spectra. This instrument makes it possible to obtain spectra ranging from 50 to 3470 cm$^{-1}$. The Raman spectra had undergone special processing, in order to be able to compare the intensity ratios between the samples. Before normalizing the spectra on the peak at 2885 cm$^{-1}$, which corresponded to the most intense peak in the spectral region 3200–50 cm$^{-1}$, or on the peak at 1743 cm$^{-1}$, which corresponded to the most intense peak in the spectral region 1800–1600 cm$^{-1}$, we subtracted the baseline for each spectrum.

### 2.6. Statistical Analysis

Regarding the determination of the Raman and ATR spectra, we chose as a research protocol to make five different spectra for each sample in order to observe the reproducibility of the results. In micro Raman, we proceeded to the determination of five spectra on five different positions of the incident laser for each sample, while in ATR-FTIR we carried out the same experiment five times by taking the same sample five times. Whether with ATR or Raman spectra, we observed no difference between the spectra for the same sample, confirming the reproducibility of our results.

In order to refine our study, we can perform models by Lorentzian functions of MIR and Raman spectra to determine the values of the wavenumbers as well as the areas of the associated peaks. The values in frequencies of the modes are obtained by the modeling carried out by the software ORIGIN 5.0 professional (from OriginLab, Northampton, MA, USA). The modeling method was proposed by Bresson et al. [16]. Statistical analysis of the data was performed by analysis of variance (ANOVA) by the software ORIGIN 5.0. The level of significance was defined as $p \leq 0.05$.

We used as peak function the following Lorentz function: $y = y_0 + \frac{2A}{\pi} \cdot \frac{w}{4(x-x_c)^2 + w^2}$ where $x_c$ represents the value of the fit mode wavenumber, $A$ the area of the peak and $w$ the width at mid-height. We took an iteration number equal to 100. The error was estimated to be $\pm 0.5$ cm$^{-1}$.

## 3. Results and Discussion

### 3.1. Polymorphic Discrimination at 20 °C

Figure 1 exhibits the intensity of SAXS (Small Angle X-ray Scattering) (Figure 1a) and WAXS (Wide Angle X-ray Scattering) (Figure 1b) configurations at room temperature for "pure" compounds, CB, CBE and AMF, and for mixtures CB + 0.5AMF, CB + AMF, CB + 0.5AMF + CBE and CB + AMF + CBE. Peaks at small angles are assigned to long d-spacings, reflecting the lamellar structure of TG; peaks at wide angles correspond to short d-spacing, defining distances between chains of TG.

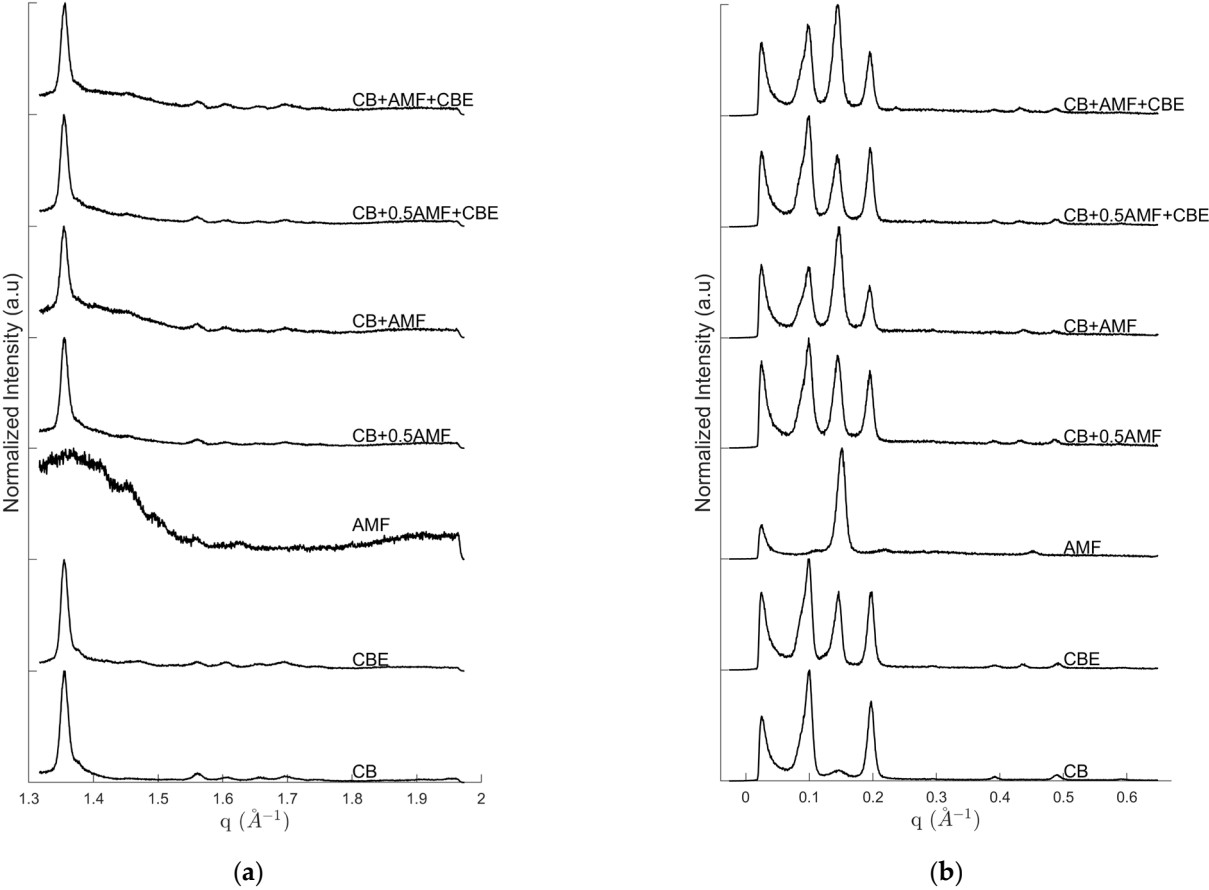

**Figure 1.** WAXS (**a**) and SAXS (**b**) diffractograms of CB, CBE, AMF, CB + AMF, CB + AMF + CBE, CB + 0.5AMF and CB + 0.5AMF + CBE, at room temperature.

### 3.1.1. "Pure" Compounds

In the case of CB, the following diffraction peaks are identified on the diffractogram: 63.5 Å (q = 0.099 Å$^{-1}$), 43.0 Å (q = 0.146 Å$^{-1}$) and 32.1 Å (q = 0.196 Å$^{-1}$) and higher orders at 16.1 Å (q = 0.391 Å$^{-1}$), and 12.8 Å (q= 0.489 Å$^{-1}$) for SAXS and 4.03 Å (q = 1.56 Å$^{-1}$), 3.90 Å (q = 1.61 Å$^{-1}$), 3.79 Å (q = 1.66 Å$^{-1}$), 3.70 Å (q = 1.70 Å$^{-1}$) and 3.59 Å (q = 1.75 Å$^{-1}$) for WAXS. CBE differs from CB because of intensity ratios, especially due to the presence of a more intense diffraction peak at 43.0 Å (q = 0.146 Å$^{-1}$) which leads also to a high order small peak at 14.3 Å (q = 0.439 Å$^{-1}$) in SAXS region and a small peak at 4.27 Å (q = 1.47 Å$^{-1}$) in the WAXS. Mainly one intense peak at 41.0 Å (q = 0.154 Å$^{-1}$) and higher order at 13.9 Å (q = 0.453 Å$^{-1}$) are visible on AMF diffractogram.

Based on studies on CB, CBE and AMF [7,13,14], the diffraction peaks can be assigned: for SAXS, the peak at 63.5 Å corresponds to the first order of the triple chain structure $3L_{001}$, the peak at 32.1 Å to the second order of the same structure ($3L_{002}$) and the peak of center at 43.0 Å to the first order of a double length chain structure $2L_{001}$. For WAXS, we can identify CB V($\beta_1$) form due to the last six peaks (4.03, 3.90, 3.79, 3.70 and 3.59 Å). The presence of the peaks at 4.03 and 3.79 Å indicates that V form is involved, and not VI form, according to the literature. Indeed, during the preparation of the mixtures, the pure compounds were melted up to T = 60 °C: the samples' fusion erased their polymorphic history. It is therefore not surprising that some TGs in the mixture are still in V form, especially when the sample has been stored under optimal conditions (at a temperature of 4 °C) as Bresson et al. [13] showed with the obtaining of the cocoa butter polymorph isolated protocol. The peak at 4.27 Å was present in the diffractogram of CBE at room temperature and was assigned to the 2L-β′ structure of POP triglycerides [14,17].

In addition, we notice a strong variation in the intensity of the peak at 43 Å compared to the other two peaks at 63 Å and 33 Å depending on the type of mixture, or compared to CB and CBE. This will lead us to consider a new parameter $\rho$, which takes into account the part of the intensity of the $I_{43}$ peak at 43 Å compared to $I_{63}$ peak in SAXS: $\rho = \frac{I_{43}}{I_{63}} = \frac{I(2L_{001})}{I(3L_{001})}$. From the study of this parameter, we will be able to propose an approximation of the proportion of TGs in β' form relative to the whole mixture. At 20 °C temperature, this parameter $\rho$ becomes worthy for CB and CBE: $\rho_{CB} = 0.127$ and $\rho_{CBE} = 0.729$. These results seem to indicate that for CBE there are 83% $\left( = \frac{\rho_{CBE} - \rho_{CB}}{\rho_{CBE}} \right)$ more TGs in IV or β' form than for CB.

Concerning AMF, SAXS shows the typical 2L long spacing around 41 Å (q = 0.154 Å$^{-1}$ and third order at 0.453 Å$^{-1}$); WAXS are noisy, certainly because the solid fraction is low at room temperature and it is not possible to see the weak b' signature described in the literature [15].

### 3.1.2. CB-AMF Mixtures without CBE

In the case of CB + 0.5AMF mixture, the following diffraction peaks are identified on the diffractogram (Figure 1): 63.5 Å (q = 0.099 Å$^{-1}$), 43.3 Å (q = 0.145 Å$^{-1}$) and 32.4 Å (q = 0.194 Å$^{-1}$), with high orders at 16.1 Å (q = 0.391 Å$^{-1}$), 14.6 Å (q = 0.431 Å$^{-1}$) and 12.9 Å (q = 0.486 Å$^{-1}$) for SAXS and 4.00 Å (q = 1.57 Å$^{-1}$), 3.89 Å (q = 1.61 Å$^{-1}$), 3.79 Å (q = 1.66 Å$^{-1}$), 3.70 Å (q = 1.70 Å$^{-1}$) and 3.59 Å (q = 1.75 Å$^{-1}$) for WAXS.

Regarding the SAXS, it is interesting to note that the 2L structure seems to be imposed by the CB as the position of its third order, which is at 0.431 Å$^{-1}$ instead of 0.453 Å$^{-1}$ for AMF. This is less discriminant for the first orders, because the peak positions are very close but the peak at 0.145 Å$^{-1}$ is similar to that of CB alone (0.146 Å$^{-1}$) and slightly different to AMF alone (0.154 Å$^{-1}$).

At T = 20 °C, the parameter $\rho$ becomes worthy for this mixture: $\rho_{CB+0.5AMF} = 0.938$. Compared to CB and CBE, this parameter is more important: it increases by 87% between CB and CB + 0.5AMF, whereas by mass TGs in β' form have increased by 50%, and by 22% between CBE and CB + 0.5AMF.

After increasing the amount of AMF in the CB + AMF mixture it can be seen that the peak positions are practically the same as for CB + 0.5AMF mixture. This is also true for 2L structure which keep the "positions of CB" even if the proportion of AMF is more important. In addition, the intensity of $2L_{001}$ peak compared to $3L_{001}$ and $3L_{002}$ peaks is much greater for "CB + AMF" mixture than for "CB + 0.5AMF" mixture.

At T = 20 °C, we measure a value for the parameter $\rho$ for CB + AMF of 1.531 while for $\rho_{CB+0.5AMF}$(T = 20 °C) = 0.936, i.e., a 38% increase of this ratio. We have to compare this parameter with the proportion by mass of TGs in IV or β' form: between CB + 0.5AMF and CB + AMF, these TGs have increased by 33%. It would seem once again that the parameter $\rho$ linked to the results obtained in X-ray for the SXAS follows the same evolution as the parameter μ linked to the mass distribution of TGs according to their polymorphic forms.

### 3.1.3. CB-AMF Mixtures with CBE

From Figure 1 the positions of the diffraction peaks were identified: 64.8 Å (q = 0.097 Å$^{-1}$), 44.2 Å (q = 0.142 Å$^{-1}$) and 32.9 Å (q = 0.191 Å$^{-1}$), with high orders at 16.1 Å (q = 0.391 Å$^{-1}$), 14.6 Å (q = 0.431 Å$^{-1}$) and 12.9 Å (q = 0.486 Å$^{-1}$); and for WAXS: 4.00 Å (q = 1.57 Å$^{-1}$), 3.89 Å (q = 1.61 Å$^{-1}$), 3.79 Å (q = 1.66 Å$^{-1}$), 3.70 Å (q = 1.70 Å$^{-1}$) and 3.59 Å (q = 1.75 Å$^{-1}$).

We find the same peaks as the last two mixtures presented. The peak assignment is therefore the same: the peaks at 64.8 Å, 44.2 Å and 32.9 Å correspond respectively to the structures $3L_{001}$, $2L_{001}$ and $3L_{002}$; and the peaks observed in WAXS correspond to $\beta_1$ form and to β' form.

At T = 20 °C, we measure a value for the parameter $\rho$ for CB + 0.5AMF + CBE of 0.694 while $\rho_{CBE}$(T = 20 °C) = 0.729, i.e., very similar values. Regarding the CB + AMF + CBE mixture, the parameter $\rho$ reaches 1.233 while $\rho_{CB+0.5AMF+CBE}$(T = 20 °C) = 0.694, i.e., a 44% increase of this ratio. For this mixture, the proportion by mass of TGs from AMF has

increased by 40% comparatively to CB + 0.5AMF + CBE. It would seem that the ρ parameter is directly related to the mass distribution of TGs in IV form at the temperature T = 20 °C.

### 3.2. Behavior with Temperature

Figures 2 and 3 deal with the thermal behavior of the mixtures. More precisely, DSC curves obtained during first heating from room temperature to 50 °C are plotted in Figure 2. For all samples, the curve is not flat at the start of the experiment, which indicates that the samples are not completely solid at room temperature. Indeed, the mixtures have a pasty structure and are quite flexible at room temperature, unlike the pure compounds which seemed solid to the touch.

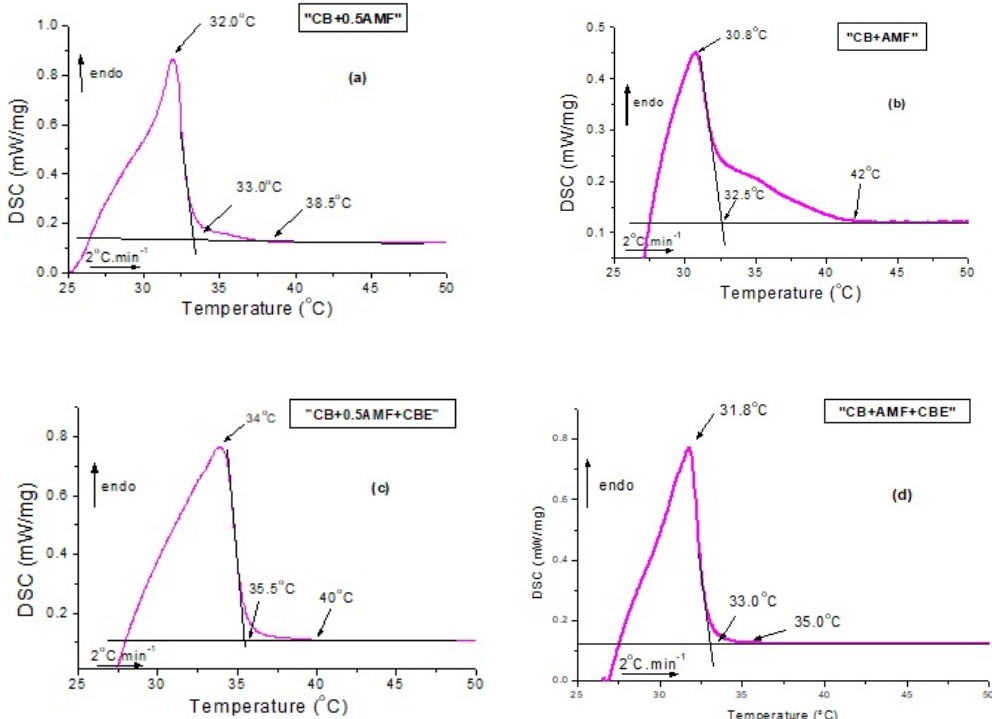

**Figure 2.** Differential scanning calorimetry (DSC) recordings measured during heating at 2 °C/min from 22 °C to 60 °C for CB + 0.5AMF (**a**), CB + AMF (**b**), CB + 0.5AMF + CBE (**c**) and CB + AMF + CBE (**d**).

Several events are clearly visible on the DSC traces with a mean peak between 30.8 °C and 34 °C (position of the peak maximum) and shoulders before (as is the case for CB + 0.5AMF, CB + 0.5AMF + CBE and CB + AMF + CBE mixtures) or after (as in the case of CB + AMF mixture). This suggests the presence of a second endothermic event of lower intensity. We deduce the coexistence of two distinct crystal structures at room temperature: one relating to the *2Lβ'* form and the other in the *3Lβ* form. This is confirmed by the SAXS patterns with temperature. For all the mixtures, the 3L structure (peaks at 64 and 32 Å) finishes to melt first, followed by the 2L organizations (peak at 44 Å) due to AMF and/or CBE.

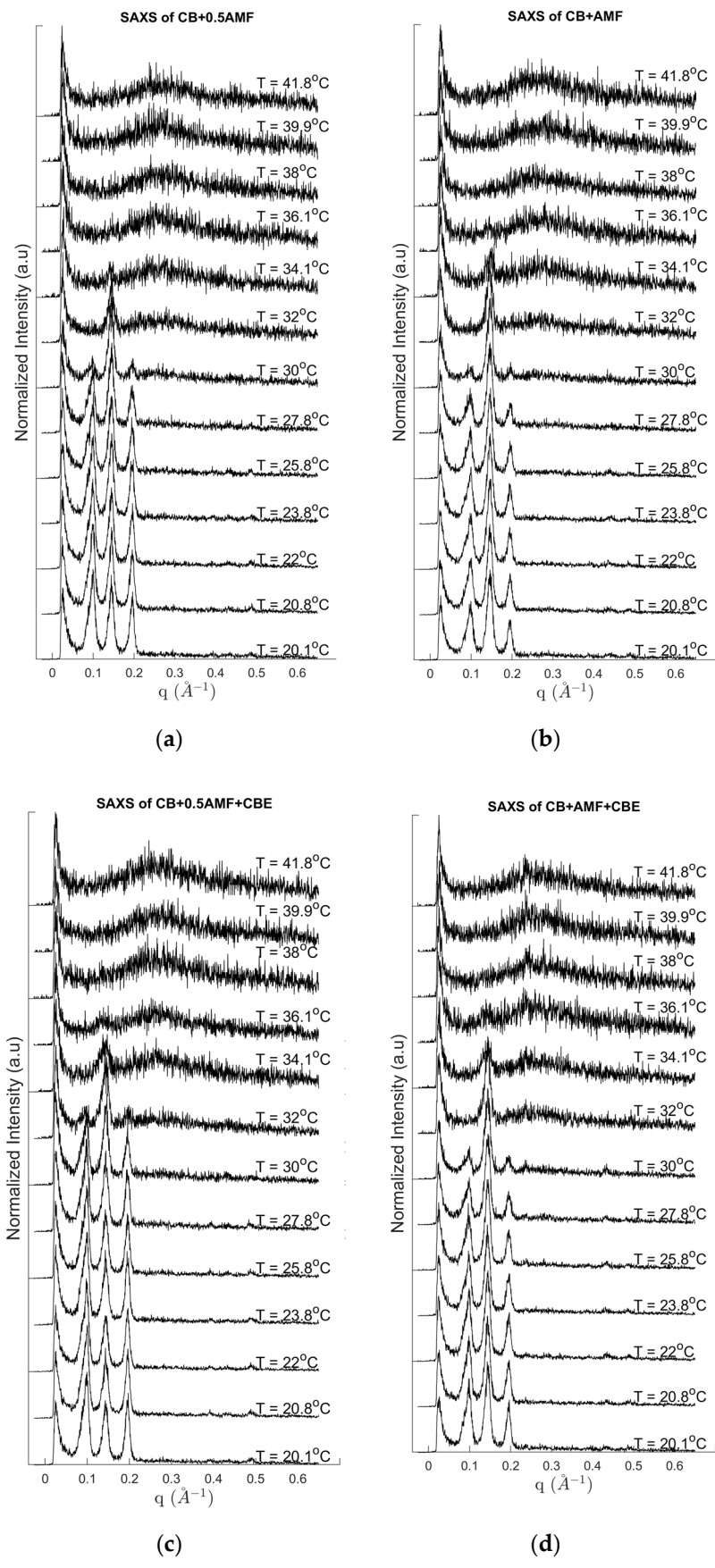

**Figure 3.** SAXS pattern recordings during heating at 2 °C/min for CB + 0.5AMF (**a**), CB + AMF (**b**), CB + 0.5AMF + CBE (**c**) and CB + AMF + CBE (**d**).

### 3.3. MIR Investigations

MIR spectra of the four mixtures in the spectral zone 3100–600 cm$^{-1}$, then the spectral zones 3020–2800 cm$^{-1}$ and 1780–1700 cm$^{-1}$ at T = 22.0 °C, are presented respectively in Figure 4a–c.

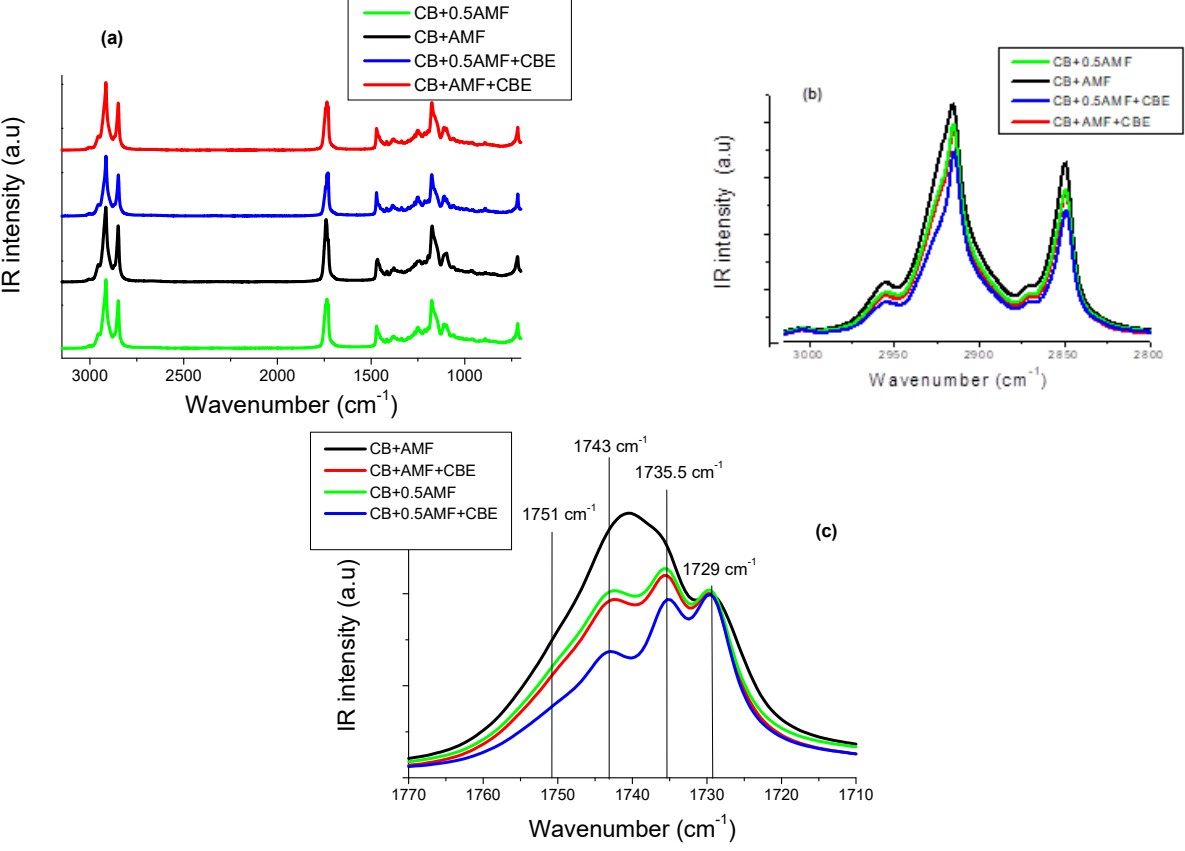

**Figure 4.** The 3100–600 cm$^{-1}$ MIR spectral range (**a**), the 3100–2800 cm$^{-1}$ MIR spectral range (**b**) and the 1770–1710 cm$^{-1}$ MIR spectral range (**c**) of the four mixtures: CB + 0.5AMF, CB + AMF, CB + 0.5AMF + CBE and CB + AMF + CBE.

Bresson et al. [6] showed that CBE exhibits the same vibrational behavior in MIR spectroscopy as CB. It is questionable whether the presence of CBE, which delays chocolate blooming, manifests on MIR spectra.

We can distinguish different regions on the spectra shown in Figure 4a. Here, we will focus on the two following regions:

- the spectral region 3200–2700 cm$^{-1}$ corresponding to the ν(C–H) stretching mode (Figure 4b)
- the spectral region 1800–1700 cm$^{-1}$ corresponding to the ν(C=O) ester carbonyl stretching region (Figure 4c).

If CBE has the same vibrational behavior in MIR spectroscopy as CB in the mixtures, MIR spectra of "CB + 0.5AMF" and "CB + AMF + CBE" mixtures should be identical. Indeed, both of these two mixtures contain one third (1/3) of AMF and two thirds (2/3) of cocoa butter (only CB or a mixture of CB and CBE).

In the spectral region 1800–1700 cm$^{-1}$ (Figure 4c), it actually seems that there is no notable difference between the spectra of "CB + 0.5AMF" and "CB + AMF + CBE" mixtures. The models of MIR spectra of the four mixtures in this spectral region are presented in Figure 4b. In Figure 4a, the spectra obtained for CB, CBE and AMF at room temperature are presented for comparison [6]. We observe four components in total, whose values of

σ remain almost stable from one mixture to another: 1729 cm$^{-1}$; 1735.5 cm$^{-1}$; 1743 cm$^{-1}$ and 1751 cm$^{-1}$.

Moreover, it should be noted that the pure compounds also presented these four components for the vibrations ν(C=O) (see Figure 5).

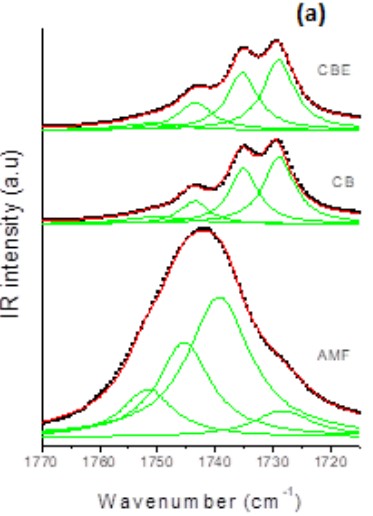 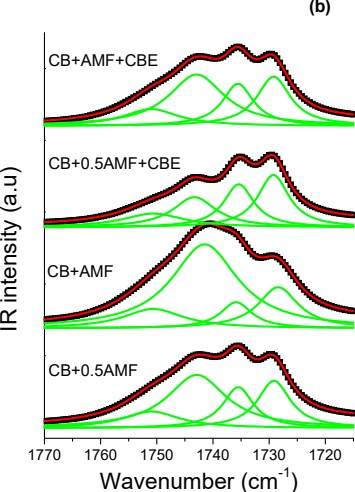

**Figure 5.** The MIR carbonyl stretching region (1770–1710 cm$^{-1}$) fitted by Lorentzian curves of CB, CBE and AMF (**a**) and of the four mixtures (**b**): CB + 0.5AMF, CB + AMF, CB + 0.5AMF + CBE and CB + AMF + CBE, at room temperature. The solid lines represent the components fitted by Lorentzian functions.

The values of the peaks of the four mixtures as well as those of the compounds alone are grouped together in Table 2. The values of the peaks in cm$^{-1}$ of the mixtures and of the pure compounds are almost identical, except for the component at 1735.5 cm$^{-1}$ for AMF which is located at 1739.8 cm$^{-1}$. In Table 2, we present the values of the areas of modes with each component. It can be seen that "CB + 0.5AMF" areas and "CB + AMF + CBE" mixtures are similar. In conclusion, it would seem that the presence of CBE instead of CB is not visible in MIR spectroscopy in the 1800–1700 cm$^{-1}$ area.

**Table 2.** Areas of the components of MIR spectra for C=O carbonyl group for CB, CBE, AMF and mixtures.

| Area (a.u) | 1729.0 cm$^{-1}$ | 1735.5 cm$^{-1}$ | 1743.0 cm$^{-1}$ | 1751.0 cm$^{-1}$ |
|:---:|:---:|:---:|:---:|:---:|
| *AMF* | 6.0 | 34.0 | 19.0 | 9.4 |
| *CB* | 8.0 | 6.2 | 2.5 | 1.8 |
| *CBE* | 8.5 | 6.8 | 3.4 | 1.4 |
| *CB + 0.5AMF* | 7.3 | 5.9 | 14.8 | 4.5 |
| *CB + AMF* | 7.7 | 3.8 | 26.2 | 5.5 |
| *CB + 0.5AMF + CBE* | 7.7 | 6.2 | 6.1 | 4.1 |
| *CB + AMF + CBE* | 7.8 | 6.2 | 13.5 | 4.4 |

In Figure 5b, it can be seen that some components have larger areas in some mixtures than in others. It is therefore interesting to comment on the evolution of the intensity and the area of each peak from one mixture to another, using the values of the areas of each component (Table 3). For example, the peak area at 1743 cm$^{-1}$ is the highest for "CB + AMF" mixture, which contains the most AMF (50%) (area = 26.2 au) and the lowest for "CB + 0.5AMF + CBE" mixture, which contains the least (20% of AMF) (area = 6.1 au). "CB + 0.5AMF" and "CB + AMF + CBE" mixtures, which both contain 33% of AMF, have an area of intermediate and almost identical values (area = 13.5 au for CB + AMF + CBE and area = 14.8 for CB + 0.5AMF). It would therefore seem that the peak at 1743 cm$^{-1}$ mainly reflects the vibrations ν(C=O) of TGs of AMF.

**Table 3.** Assignment of the elongation vibrations ν(CH) of MIR spectrum of the four mixtures at room temperature in the 3200–2800 cm$^{-1}$ zone. Legend: m = medium, s = strong; vs = very strong; sh = shoulder; ν = stretch; s = symmetric; as = asymmetric.

| Wavenumber cm$^{-1}$ | Intensity | Assignation |
|:---:|:---:|:---:|
| 2849 | (s) | $\nu_s(CH_2)$ |
| 2852 | (sh) | $\nu_s(CH_2)$ |
| 2872 | (sh) | $\nu_s(CH_2)$ |
| 2902 | (sh) | $\nu_{as}(CH_2)$ |
| 2915 | (s) | $\nu_{as}(CH_2)/\nu_s(CH_3)/\nu_{as}(CH_3)$ |
| 2926 | (sh) | $\nu_{as}(CH_2)$ |
| 2957 | (m) | $\nu_{as}(CH_3)$ |
| 3005 | (vs) | $\nu(CH)$ unsaturated |

We observe an identical phenomenon for the peak at 1751 cm$^{-1}$: its area is the most intense for "CB + AMF" mixture (area = 5.5 au), and the least intense for "CB + 0.5AMF + CBE" mixture (area = 4.1 au). For the other two mixtures containing the same percentage by mass of AMF, we observe similar areas for the peak at 1751 cm$^{-1}$: area ≈ 4.5 a.u. Thus, it would seem that the peak at 1751 cm$^{-1}$ would be represented over all the vibrations ν(C=O) of TGs of AMF. These results are consistent in comparison with the area of the peaks at 1751 cm$^{-1}$ and 1743 cm$^{-1}$ between CB, CBE and AMF taken individually: the areas of these two peaks are very important for AMF taken alone (respectively 9.4 and 19.0 au) unlike CB and CBE (respectively 1.8 and 2.5 au for the CB and 1.4 and 3.4 for CBE) (see Table 3). The peak at 1735.5 cm$^{-1}$ has an almost identical area from one mixture to another, except for "CB + AMF" mixture where the area is lower. However, it is the mixture which contains the least cocoa butter (equivalent or not): 50% of (CB + CBE), against 67% for "CB + 0.5AMF" and "CB + AMF + CBE" mixtures and 80% for "CB + 0.5AMF + CBE" mixture. It would therefore seem that the peak at 1735.5 cm$^{-1}$ would mostly represent the vibrations ν(C = 0) of TGs of CB and CBE. Since the peak at 1729 cm$^{-1}$ was used to carry out the normalization, we will not be able to comment on its evolution.

In the 3200–2700 cm$^{-1}$ zone corresponding to the elongation vibration ν(C–H) of TGs, MIR spectra of the four mixtures are shown in Figure 4b. In Figure 6, the modeling of MIR spectrum of "CB + 0.5AMF + CBE" mixture is presented by Lorentzian functions with the same modeling method used for the spectral zone 1800–1700 cm$^{-1}$. If we apply this modeling method to each of the spectra of the four mixtures, we find that all the mixtures have the same eight components: a very weak peak at 3005 cm$^{-1}$; a defined but weak peak at 2957 cm$^{-1}$; a fine and intense peak at 2915 cm$^{-1}$ with two shoulders on either side, at 2926 and 2902 cm$^{-1}$; a very weak peak of intensity at 2872 cm$^{-1}$ and then a fine and intense peak at 2849 cm$^{-1}$ with a shoulder at 2852 cm$^{-1}$. We find the same peaks observed for CB, CBE [14] and the AMF. The assignments of these peaks are shown in Table 3 [18]. From Figure 4b, it seems that we do not observe a difference in vibrational behavior between the mixtures for this spectral zone.

In conclusion, CBE TGs have the same vibrational behavior in MIR spectroscopy as CB TGs in mixtures. This is not very surprising, since the study of CB alone and CBE alone did not reveal any vibrational differences in MIR [14]. On the other hand, the detailed study of spectra in MIR spectroscopy in the spectral zone 1800–1700 cm$^{-1}$ allows us to know the major contribution of one of the elements of the mixtures for three vibrational modes: the peaks at 1751 and 1743 cm$^{-1}$ are predominantly sensitive to AMF while the peak at 1735.5 cm$^{-1}$ is sensitive to CB and CBE.

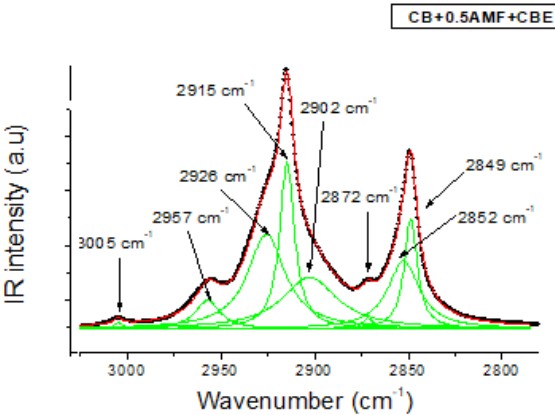

**Figure 6.** The MIR C-H stretching region (3050–27500 cm$^{-1}$) fitted by Lorentzian curves of CB + 0.5AMF + CBE at room temperature. The solid lines represent the components fitted by Lorentzian functions.

### 3.4. Raman Investigations

Bresson et al. [14] showed that MIR spectroscopy study of CB and CBE failed to find vibrational modes that could differentiate CB from CBE, unlike Raman spectroscopy. In addition, A. Lambert et al. showed a vibrational behavior for AMF quite different from CB and CBE in Raman spectroscopy at room temperature [15]. Raman spectra at room temperature for the four mixtures are represented in Figure 7a, 7b and 7c respectively in the spectral zones 3100–600, 3100–2700 and 1770–1710 cm$^{-1}$.

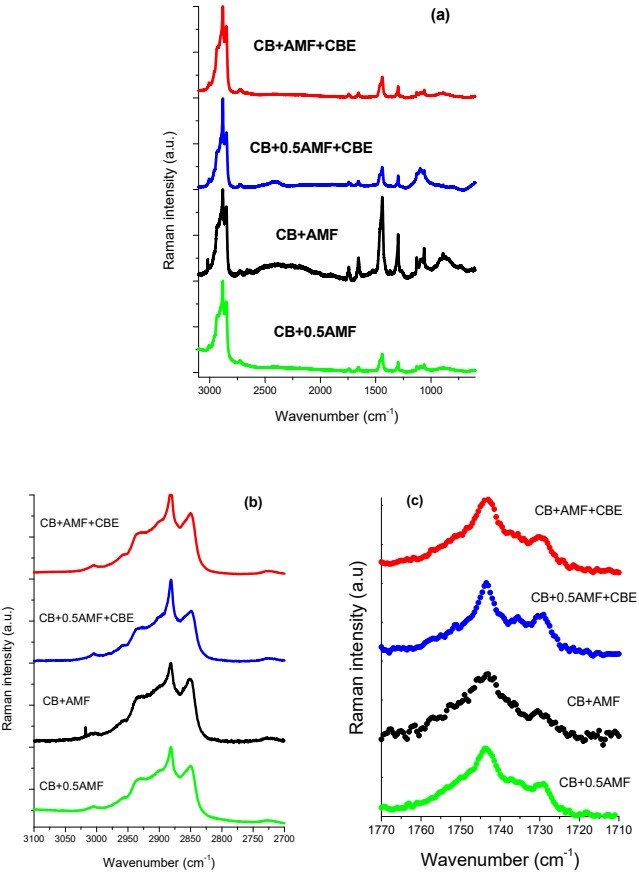

**Figure 7.** The 3100–600 cm$^{-1}$ Raman spectral range (**a**), the 3100–2800 cm$^{-1}$ Raman spectral range (**b**) and the 1770–1710 cm$^{-1}$ Raman spectral range (**c**) of the four mixtures: CB + 0.5AMF, CB + AMF, CB + 0.5AMF + CBE and CB + AMF + CBE.

The spectral region 1800–1700 cm$^{-1}$ corresponding to the vibration of elongation of the carbon-oxygen double bonds, noted $\nu$(C=O), was found to be very rich for the three pure compounds [14,15]. It is therefore interesting to study this zone for mixtures. For this, one carries out modeling by Lorentzian functions for the four mixtures (see Figure 8b). In order to analyze this spectral region more easily, Raman spectra of the pure compounds at room temperature are presented (see Figure 8a). Modeling makes it possible to determine the value in cm$^{-1}$ of the center of the peaks (Table 4) as well as the areas of each peak (Table 5).

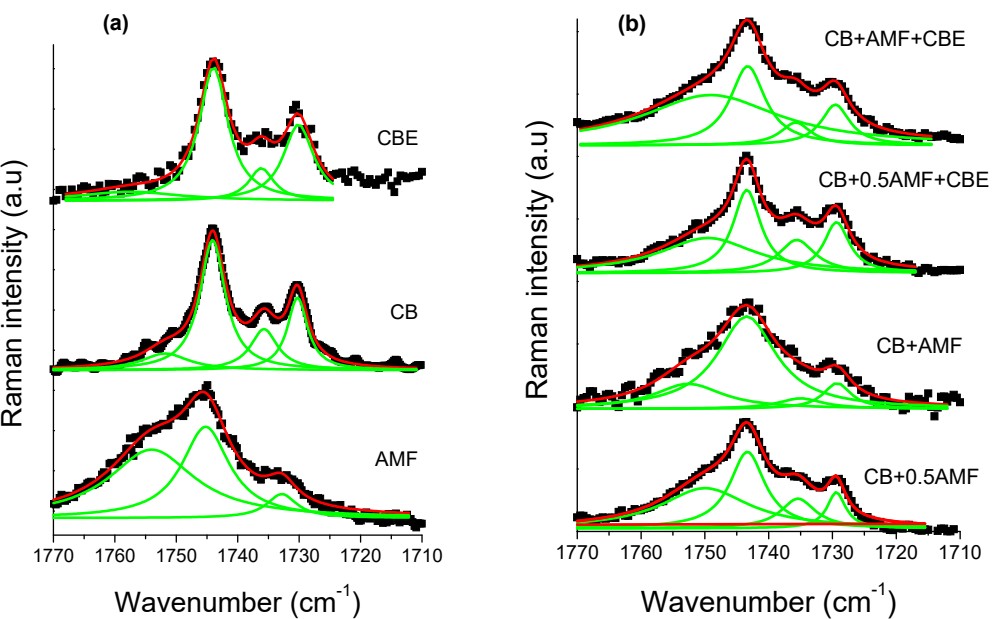

**Figure 8.** The Raman carbonyl stretching region (1770–1710 cm$^{-1}$) fitted by Lorentzian curves of CB, CBE and AMF (**a**) and of the four mixtures (**b**): CB + 0.5AMF, CB + AMF, CB + 0.5AMF + CBE and CB + AMF + CBE, at room temperature. The solid lines represent the components fitted by Lorentzian functions.

**Table 4.** Values of the $\nu$(C=O) (cm$^{-1}$) of the peaks of Raman spectra of CB, CBE, AMF and mixtures at room temperature. * values obtained by Lorentzian modeling.

| Sample | $v(C=0)$ (cm$^{-1}$) | $v(C=0)$ (cm$^{-1}$) | $v(C=0)$ (cm$^{-1}$) | $v(C=0)$ (cm$^{-1}$) |
|---|---|---|---|---|
| **CB** | 1751 * | 1744 | 1736 | 1730 |
| **CBE** | 1755 * | 1743 | 1736 | 1730 |
| **AMF** | 1754 * | 1745 | *no peak* | 1732 |
| **CB + 0.5AF** | 1750 * | 1743 | 1736 | 1730 |
| **CB + AMF** | 1751 * | 1743 | 1735 * | 1729 |
| **CB + 0.5AMF + CBE** | 1750 * | 1743 | 1735 | 1729 |
| **CB + AMF + CBE** | 1750 * | 1743 | 1735 | 1729 |

**Table 5.** Evolution of the area ratios of the components of Raman spectra for the carbonyl group C=O for the four mixtures.

| Ratio of Area | CB + 0.5AMF | CB + AMF | CB + 0.5AMF + CBE | CB + AMF + CBE |
|---|---|---|---|---|
| $\tau_1 = Area_{1743}/Area_{1735}$ | 2.3 | 11.7 | 1.9 | 4.2 |
| $\tau_2 = Area_{1743}/Area_{1730}$ | 4.0 | 2.5 | 1.8 | 2.8 |
| $\tau_3 = Area_{1735}/Area_{1730}$ | 1.5 | 0.2 | 0.9 | 0.7 |

The component at 1735 cm$^{-1}$ does not exist in AMF, whereas it exists in all other samples [13]. It can thus be used as a witness of the contribution of AMF in mixtures. In Table 1, it can be seen that between "CB + 0.5AMF" and "CB + AMF" mixtures, the variations in the area ratios are very significant. The modes at 1743 and 1730 cm$^{-1}$ are common to all mixtures. Between these two mixtures, AMF's contribution is doubled. We notice that the ratio $\tau_1 = Area_{1743}/Area_{1735}$ is multiplied by four, which means that the contribution of the mode at 1743 cm$^{-1}$ has very strongly increased compared to that of the mode 1735 cm$^{-1}$. We also note that the ratio $\tau_2 = Area_{1743}/Area_{1730}$ was divided by approximately two by the contribution of AMF in "CB + AMF" mixture. This means that the contribution of the mode at 1730 cm$^{-1}$ has doubled compared to the mode at 1743 cm$^{-1}$. If we consider the ratio $\tau_3 = Area_{1735}/Area_{1730}$, we see that the addition of AMF in "CB + AMF" mixture was accompanied by a very favorable area ratio in the 1730 cm$^{-1}$ mode compared to that of the mode at 1735 cm$^{-1}$ (multiplied by 10). In view of the variation of these three rates, we can say that the addition of AMF in the mixture based on CB (we go from 33% by mass to 50% for CB + AMF mixture) was accompanied by a very strong contribution of the mode at 1730 cm$^{-1}$, then to a lesser extent of the mode contribution at 1743 cm$^{-1}$ and finally to a small extent of the mode at 1735 cm$^{-1}$.

When CBE is added to the two previous mixtures, the influence of AMF on the variation of the different area ratios is less pronounced than without CBE addition. The ratio $\tau_1$ is multiplied by two between CB + 0.5AMF + CBE and CB + AMF + CBE mixtures, while for CB + 0.5AMF and CB + AMF mixtures it is multiplied by four. This can be explained by a lower proportion of mass of AMF in the two new mixtures. Between "CB + 0.5AMF" and "CB + AMF" mixtures, we shift from 33% of AMF in the mixture to 50%, while for "CB + 0.5AMF + CBE" and "CB + AMF + CBE "mixtures we go from 20% to 33% by mass for AMF. However, in "CB + 0.5AMF" and "CB + AMF + CBE" mixtures where the proportion of AMF is the same, the area ratios are different. This means that the contribution of CBE in the mixture is not the same as the contribution of CB. If we compare the variations of "CB + 0.5AMF" mixtures with "CB + AMF + CBE" mixtures, we notice that $\tau_1$ has been multiplied by approximately two and $\tau_3$ has been divided by two. CBE therefore has a direct impact on the environment close to the carbonyl group in the whole crystal structure of the mixture due to the fact that some of its TGs are in IV form, while for CB they are all in V or VI form.

In the 3000 cm$^{-1}$ zone, the previous study of Raman spectra of AMF as a function of temperature highlighted a degree of crystallization marker: the intensity ratio $r = I[\nu_s(CH2)]/I[\nu_{as}(CH2)] = I_{2845}/I_{2880}$ [15]. It was observed that the intensity of the asymmetric valence mode of vibration $\nu_{as}(CH_2)$ at 2885 cm$^{-1}$ increased with respect to the intensity of the symmetric valence mode of vibration $\nu_s(CH_2)$ at 2850 cm$^{-1}$ as the temperature decreased. On the other hand, Bresson et al. demonstrated that this ratio was very sensitive to the conformation of TGs in the crystal structure of cocoa butter [13] between the liquid form of CB and its form VI. This ratio goes from 1.72 to 0.44. Table 6 gives the value of this intensity ratio for the different mixtures at room temperature.

**Table 6.** Area ratio of 2850 and 2885 cm$^{-1}$ vibrational modes of Raman spectra for CB, CBE, AMF and mixtures at room temperature.

| Sample | $r= Area_{2850}/Area_{2885}$ |
|:---:|:---:|
| *CB* | 0.42 |
| *CBE* | 0.68 |
| *AMF* | 1.10 |
| *CB + 0.5AMF* | 0.79 |
| *CB + AMF* | 0.82 |
| *CB + 0.5AMF + CBE* | 0.64 |
| *CB + AMF + CBE* | 0.80 |

From Table 6, it is noted that at room temperature, CBE is less crystallized than CB because part of its TGs is in IV form. It is also noted that the ratio remains practically identical between CB + 0.5AMF and CB + AMF + CBE mixtures, and the percentage by mass of AMF remains identical. Between CB + 0.5AMF + CBE and CB + AMF + CBE mixtures, where the percentage by mass of AMF changes from 20% to 33%, we also observe an increase in this rate from 0.64 to 0.80, while for CB and CBE alone the ratio *r* is very different. Concerning mixtures between CB, AMF and CBE, it would seem that the proportion of AMF is a determining factor in the increase or not of this report.

## 4. Conclusions

The purpose of this work was to try to understand why among four mixtures (CB + 0.5AMF, CB + AMF, CB + 0.5AMF + CBE and CB + AMF + CBE) the last two mixtures delayed the arrival of bloom chocolate by 20 weeks for CB + 0.5AMF + CBE and by 30 weeks for CB + AMF + CBE. DSC. X-ray diffraction investigations were performed for a structural study of four mixtures CB + 0.5AMF, CB + AMF, CB + 0.5AMF + CBE and CB + AMF + CBE at room temperature. They showed that, at room temperature, the crystallized TGs of CB were all in V form ($3L\beta_1$), but that part of TGs of CBE was in IV form ($2L\beta'_2$), as well as TGs from AMF. Measuring the intensity ratio $\rho = \frac{I(2L_{001})}{I(3L_{001})}$ seems to be an indicator of the number of TG form IV inside the mixture. The study of this parameter $\rho$ seems to be a very relevant indicator in determining the proportion of TG in IV form from CBE and is directly correlated with the onset time of chocolate blooming.

The study in Raman spectroscopy of the mixtures at room temperature was found to be more relevant than that in infrared spectroscopy. In the spectral region of the carbonyl group (1800–1700 cm$^{-1}$), we observed four components for CB, CBE and the mixtures, but three components for AMF: the peak at 1735 cm$^{-1}$ does not appear for AMF. The study of three ratios of areas with respect to the three components 1743, 1735 and 1730 cm$^{-1}$ ($\tau_1 = Area_{1743}/Area_{1735}$, $\tau_2 = Area_{1743}/Area_{1730}$, $\tau_3 = Aera_{1735}/Aera_{1730}$) showed strong variations between the mixtures. Thus, we found many differentiation markers showing the impact of CBE in the mixture, as Sonvaï and Rousseau showed in their study of fat bloom based on these mixtures. However, it is more difficult to quantify the contribution of Form IV TGs from CBE in the mixture by Raman spectroscopy than in the X-ray study in chocolate blooming onset time study. This study is a proof that all these techniques are complementary to realize the structural and vibrational investigation of the mixtures CB, AMF and CBE and are very relevant in the understanding of chocolate blooming.

**Author Contributions:** M.E.H.: Conceptualization, investigation and writing; S.B.: Methodology, investigation and writing; A.L.: Methodology, investigation and writing; F.B.: Methodology, investigation and writing; V.B.: Methodology and writing; V.H.N.: Methodology, investigation and writing; V.F.: Methodology, investigation and writing; M.C.: Investigation. All authors have read and agreed to the published version of the manuscript.

**Funding:** This research received no external funding.

**Institutional Review Board Statement:** Not applicable.

**Informed Consent Statement:** Not applicable.

**Data Availability Statement:** Data are contained within the article.

**Acknowledgments:** We would like to thank Quentin Arnould, technician of *Walloon Agricultural Research Centre (CRA-W)*, who participated in Raman and MIR measurements, and Jean-Jacques Vachon, *Institut Galien Paris-Saclay*, for his help during SAXS-WAXS measurements.

**Conflicts of Interest:** The authors declare no conflict of interest.

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
