# Peer review of "Structural and Vibrational Investigations of Mixtures of Cocoa Butter (CB), Cocoa Butter Equivalent (CBE) and Anhydrous Milk Fat (AMF) to Understand Fat Bloom Process"

_applsci, doi:10.3390/app12136594_

Round 1

Reviewer 1 Report

Dear Authors,

The presented manuscript aimed to carry out of structural and vibrational investigations of cocoa butter(CB), cocoa butter equivalent (CBE) and anhydrous milk fat (AMF) mixtures to understand bloom chocolate process.

 The above article perfectly fits into the thematic  framework of Special Issue entitled "New Trends in the Structure Characterization of Food" in the journal Applied Sciences.

The biggest objections to this work are the lack of statistical analysis and  discussion of the obtained results and also the use  in some cases a very old references.

However, I have some suggestions to the Authors, which help them to improve this manuscript:

  • After word Introduction, please delete the colon,
  • The used English language needs to be improved by native speaker,
  • Please delete dot after main title of article,
  • Please rewrite this section used a new references,
  • The lack a number of lines in the text of manuscript makes it difficult to identify specific notes,
  • In English articles, we do not use commas in the figures, only periods.
  • Subsections: 2.2-2.6, lack of given a research methodology,
  • How many repetitions of various analysis were performed?
  • Lack of very important subsection: Statistical Analysis. Please don’t forget complete it.
  • Figure 4a, 4b: a lack of units on X-axis.
  • Tables: 5, 6: the  lack of uniform presentation,
  • A very monotonous way of presenting of obtained results in the section Results,
  •  Please, shorten the Conclusions section and indicate the main conclusions of the study.
  • The formulation of the final conclusions will facilitate a prior statistical analysis,
  • Please, delete from the section Conclusions , the reference [1]. This is not the right place to discuss the obtained results with regard to other authors,
  • The presented article does not comply with the requirements of the Applied Sciences  editorial board: (https://www.mdpi.com/journal/applsci/instructions#preparation):
  • Lack of such sections as:  Author Contributions, Data Availability Statement or Conflicts of Interest,
  • In the section References is the lack of digital object identifier (DOI) for all references where available,

From my standpoint, this manuscript might be appropriate for publication in Journal – Applied Sciences only  after major revision,  given the above aspects.  

Author Response

Dear reviewer

We brought the following corrections:

  • After word Introduction, please delete the colon, The correction has been made.
  • Please delete dot after main title of article, The correction has been made in the new version.
  • Please rewrite this section used a new references, we have added new references in the introduction 
  • Subsections: 2.2-2.6, lack of given a research methodology,
  • How many repetitions of various analysis were performed?
  • Lack of very important subsection: Statistical Analysis. Please don’t forget complete it. All these points have been corrected in the new version
  •  Please, shorten the Conclusions section and indicate the main conclusions of the study. We shortened the conclusion 
  • The presented article does not comply with the requirements of the Applied Sciences  editorial board: (https://www.mdpi.com/journal/applsci/instructions#preparation):
  • Lack of such sections as:  Author ContributionsData Availability Statement or Conflicts of Interest, we have added these sections in the new version

Best regards,

serge Bresson

Reviewer 2 Report

This manuscript examines different methods of investigating chocolate blooming. 

The results are interesting and could have potential significance for the industry. However, some minor points have to be addressed before publishing.

The authors should add statistical processing. Please mention the number of replications of the measurements and provide standard deviations wherever is missing. 

The methods and materials section should mention MIR and Raman spectral processing. Please provide, manipulation of spectra, i.e. smoothing, baseline correction or any other processing was performed. In section 3.3 the phrase " the spectra were normalized against the peak at 1729 cm-1.
The resolution of the spectra is 1
cm-1 " should be moved in methods and materials. Why did the authors choose to normalize at 1720 cm-1?

The phrase " In order to refine our study, we can perform models by Lorentzian functions of
MIR spectra to determine the values of the wavenumbers as well as the areas of the
associated peaks. The values in frequencies of the modes are obtained by the modelizations carried out by the software ORIGIN. The modeling method is proposed by
Bresson
et al [10]. The error was estimated to be ± 0.5cm-1.  " should also be moved in the materials and methods section.

Please add the peak attributions to specific vibrations along with references. 

Please explain better the ways that these peaks yielded from curve fitting may be linked to different lipid structures of the samples. 

Author Response

Dear reviewer,

We brought the following corrections:

"The methods and materials section should mention MIR and Raman spectral processing. Please provide, manipulation of spectra, i.e. smoothing, baseline correction or any other processing was performed. In section 3.3 the phrase " the spectra were normalized against the peak at 1729 cm-1.
The resolution of the spectra is 1
cm-1 " should be moved in methods and materials. Why did the authors choose to normalize at 1720 cm-1?"

We have answered these questions by making the corrections to paragraphs 2.4  page 4 et 2.5 page 5 by using the color red for writing

"The phrase " In order to refine our study, we can perform models by Lorentzian functions of
MIR spectra to determine the values of the wavenumbers as well as the areas of the
associated peaks. The values in frequencies of the modes are obtained by the modelizations carried out by the software ORIGIN. The modeling method is proposed by
Bresson 
et al [10]. The error was estimated to be ± 0.5cm-1 " should also be moved in the materials and methods section."

These remarks have been incorporated into the statistical analysis

"Please add the peak attributions to specific vibrations along with references."

This article is an extension of two others articles (Lambert A, BougriouaF., Abbas O., Courty M., ElMarssi M., Faivre V., and Bresson. Temperature dependent Raman and X-Ray diffraction studies of anhydrousmilk fats. Food Chemistry, 2018, 267, 187 -195 and Bresson S, Lecuelle ., A., Bougrioua F., El Hadri M., Baeten V., Courty M., Pilard S., Rigaud S., Faivre V. “Comparative structural and vibrational investigations between cocoa butter (CB) and cocoa butter equivalent (CBE) by ESI/MALDI-HRMS, XRD, DSC, ATR/FTIR and Raman spectroscopy”. Food Chemistry, 2021, 363, 130319) we have written which we refer and in which all the peaks have already  been referenced and assigned

"Please explain better the ways that these peaks yielded from curve fitting may be linked to different lipid structures of the samples.

The Raman response of a sample is determined by Lorentzian functions.  All the adjusted components have a real existence here. However, their mathematical properties (intensity, width at mid-height, area) correspond to relative and not absolute values. This is why we use area or intensity ratios between peaks of the same spectrum in order to evaluate the contribution of each of the components in the total envelope of the observed modes.

Best regards,

serge Bresson

Round 2

Reviewer 1 Report

Dear Authors,

The lack a number of lines in the text of manuscript makes still  it difficult to identify specific notes.

Few previous notes are still not included in the new version of the manuscript:

  • In English articles, we do not use commas in the figures, only periods (Figure 2)
  • The presented article still does not comply with the requirements of the Applied Sciences editorial board (https://www.mdpi.com/journal/applsci/instructions#preparation):
  • Lack of such sections as:  Data Availability Statement,
  • In the section References is the lack of digital object identifier (DOI) for all references where available,
  • Please, shorten the Conclusions section and indicate the main conclusions of the study. This part of article must be rewritten,
  • Subsections: 2.2-2.6, lack of given a research methodology (lack of suitable references, were completed only descriptions),
  • Figure 4 must be shown as : Figure 4 (a); 4 (b); 4 (c).

From my standpoint, this manuscript in current version still isn’t  appropriate for publication in journal Applied Sciences, given the above aspects.  It can considered for publication in this journal only  after major revision.  

Author Response

Dear reviewer,

We brought the following corrections

  • In English articles, we do not use commas in the figures, only periods (Figure 2), we have corrected in the figure
  • The presented article still does not comply with the requirements of the Applied Sciences editorial board (https://www.mdpi.com/journal/applsci/instructions#preparation):
  • Lack of such sections as:  Data Availability Statement, we have added it to the article
  • In the section References is the lack of digital object identifier (DOI) for all references where available, we have added it to the article
  • Please, shorten the Conclusions section and indicate the main conclusions of the study. This part of article must be rewritten, the conclusion has already been reduced

Best regards,

Serge Bresson

Reviewer 2 Report

The authors have improved the manuscript. Please check if  2000-50 cm-1 is correct in materials and methods section regarding MIR measurements.

Author Response

Thank you

Round 3

Reviewer 1 Report

Dear Authors,

1.       Few previous notes are still not included in the new version of the manuscript:

     In English articles, we do not use commas in the figures and tables, only periods (Figure 2; Table 6)

     The Conclusions section must be rewritten. Please, indicate the main conclusion of the study. 

     Subsections: 2.2-2.5, lack of given a research methodology (lack of suitable references, were completed only descriptions).

2.       In what option of format offered by publisher MDPI  was the manuscript prepared?

 https://www.mdpi.com/journal/applsci/instructions#preparation

 3.       Page 13: Why three tables with different data are presented as one Table 6?

The presented manuscript, even though it is the next, improved version, unfortunately still has a lot of technical flaws.

From my standpoint, this article in current version still isn’t  appropriate for publication in journal Applied Sciences, given the above aspects.  It can considered for publication in this journal again only  after major revision, based on the above comment. 

Author Response

Dear reviewer,
You are right our article needed more corrections. Indeed, we replaced the commas on figure 2 and table 6.
I apologize for that. I had not seen them.
we refocused our conclusion on the bloom chocolate by showing several measurement parameters
in X-ray diffraction and Raman spectroscopy. W
e hope you will be satisfied.
Regarding your question regarding the format of the manuscript, we used the word format. We have provided more details in the DSC part and sections 2.2 to 2.5.

Best regards,
Serge Bresson

Round 4

Reviewer 1 Report

Dear Authors,

Majority of previous comments have been taken into account.

I have only two minor comments to the revised text of the manuscript:

     Subsections: 2.2-2.5, lack of given a research methodology (lack of suitable references, still are completed only descriptions),

·                   Figure 5: Please, replace indicator of figure “a” to the right side, the same as is presented part “b”.

I don’t have more  objections to the new version of the manuscript.

From my point of view the new version of the article is appropriate for publication in Journal – Applied Sciences,  based on the above comment, after minor revision.

Author Response

Dear Reviewer,
I completed the statistical analysis part. I hope this will suit you.
I also changed figure 5.

Best regards
serge Bresson
